# Bioactive Alkaloids from the Mangrove-Derived Fungus *Nigrospora oryzae* SYSU-MS0024

**DOI:** 10.3390/md22050214

**Published:** 2024-05-09

**Authors:** Xiaokun Chen, Senhua Chen, Heng Guo, Xin Lu, Hongjie Shen, Lan Liu, Li Wang, Bin Chen, Yi Zhang, Yayue Liu

**Affiliations:** 1Guangdong Provincial Key Laboratory of Aquatic Product Processing and Safety, Guangdong Province Engineering Laboratory for Marine Biological Products, Guangdong Provincial Engineering Technology Research Center of Seafood, Key Laboratory of Advanced Processing of Aquatic Product of Guangdong Higher Education Institution, Zhanjiang Municipal Key Laboratory of Marine Drugs and Nutrition for Brain Health, Research Institute for Marine Drugs and Nutrition, College of Food Science and Technology, Guangdong Ocean University, Zhanjiang 524088, China; 13129717040@163.com (X.C.); wangli991003@163.com (L.W.); hubeizhangyi@163.com (Y.Z.); 2School of Marine Sciences, Sun Yat-sen University, Zhuhai 519000, China; chensenh@mail.sysu.edu.cn (S.C.); hengeguo163@163.com (H.G.); luxin36@mail2.sysu.edu.cn (X.L.); cesllan@mail.sysu.edu.cn (L.L.); 3Southern Laboratory of Ocean Science and Engineering (Guangdong, Zhuhai), Zhuhai 519000, China; shenhj5@mail2.sysu.edu.cn (H.S.); chenbin@sml-zhuhai.cn (B.C.); 4Collaborative Innovation Center of Seafood Deep Processing, Dalian Polytechnic University, Dalian 116034, China

**Keywords:** alkaloid, mangrove fungus, *Nigrospora oryzae*, anti-neuroinflammatory activities, AChE inhibitory activities

## Abstract

Chemical investigation of marine fungus *Nigrospora oryzae* SYSU-MS0024 cultured on solid-rice medium led to the isolation of three new alkaloids, including a pair of epimers, nigrosporines A (**1**) and B (**2**), and a pair of enantiomers, (+)-nigrosporine C (+)-**3**, and (−)-nigrosporine C (−)-**3**, together with eight known compounds (**4**–**11**). Their structures were elucidated based on extensive mass spectrometry (MS) and 1D/2D nuclear magnetic resonance (NMR) spectroscopic analyses and compared with data in the literature. The absolute configurations of compounds **1**–**3** were determined by a combination of electronic circular dichroism (ECD) calculations, Mosher’s method, and X-ray single-crystal diffraction technique using Cu Kα radiation. In bioassays, compound **2** exhibited moderate inhibition on NO accumulation induced by lipopolysaccharide (LPS) on BV-2 cells in a dose-dependent manner at 20, 50, and 100 μmol/L and without cytotoxicity in a concentration of 100.0 μmol/L. Moreover, compound **2** also showed moderate acetylcholinesterase (AChE) inhibitory activities with IC_50_ values of 103.7 μmol/L. Compound **5** exhibited moderate antioxidant activity with EC_50_ values of 167.0 μmol/L.

## 1. Introduction

Mangrove fungi have shown promising potential as sources of structurally novel and bioactive natural products because of the chemical diversity of their secondary metabolites [1,2]. To date, more than 1500 new metabolites have been found from Mangrove-derived fungi [1,3,4,5,6]. Among them, alkaloids are a kind of important metabolites with chemical structural diversity and pharmacological activity [1,7,8], such as cytotoxic [9], antibacterial [10], antiviral [11], anti-inflammatory [12], and enzyme inhibitor activities [13]. For example, epipolythiodioxopiperazine (ETP) alkaloid, penicisulfuranol A, was isolated from the mangrove endophytic fungus *Penicillium janthinellum* HDN13-309, which was a C-terminal inhibitor of Hsp90 showing cytotoxicity [14], and pyrazinopyrimidine-type alkaloid, deg-pyrasplorine B, showed anti-influenza A virus H1N1 activity with the IC_50_ of 50 μmol/L [15].

The mangrove-derived fungus *Nigrospora oryzae* SYSU-MS0024 was isolated from sediment, which was collected from the Zhuhai Mangrove Special Protected Area, Guangdong Province, China. Chemical investigation of its EtOAc extract led to the discovery of three new alkaloids, including a pair of epimers, nigrosporines A (**1**) and B (**2**), and a pair of enantiomers, (+)-nigrosporine C and (−)-nigrosporine C (**3**), together with eight known compounds (**4**–**11**) (Figure 1). Herein, the details of the isolation, structural elucidation, and bioactivities of compounds **1**–**11** are reported.

## 2. Results and Discussion

The EtOAc extract of mangrove sediment-derived fungus *Nigrospora oryzae* SYSU-MS0024 was performed on repeated silica gel and Sephadex LH-20 column chromatography, followed by semi-preparative HPLC to afford three new alkaloids, nigrosporines A–C (**1**–**3**), and eight known compounds **4**–**11**.

Nigrosporine A (**1**) was obtained as a brown crystal. The molecular formula was determined as C_13_H_15_NO_3_ on the basis of the positive HR-ESIMS ions at *m*/*z* 256.0945 [M + Na]^+^ (calculated for 256.0950, C_13_H_15_NO_3_Na), implying seven degrees of unsaturation. The IR spectrum of **1** revealed the presence of a hydroxy (3365 cm^−1^) group. The ^1^H NMR spectrum (Table 1) of **1** exhibited the signals for four aromatic protons owing to a 1,2-disubstituted aromatic ring [*δ*_H_ 7.30 (1H, t, *J* = 6.6 Hz); 7.14 (1H, td, *J* = 7.5, 1.2 Hz); 7.05 (1H, d, *J* = 7.8 Hz); 8.14 (1H, d, *J* = 8.4 Hz)], a methyl (*δ*_H_ 1.13, *J* = 6.4 Hz), three methylenes (*δ*_H_ 2.62, m; 2.62, m, 1.98, m; 5.04, d, *J* = 15.6 Hz, 4.92, d, *J* = 15.6 Hz), and a methine (4.22, qd, *J* = 6.4, 1.1 Hz). The ^13^C NMR and HSQC data (Table 1) of **1** revealed the presence of 13 carbons belonging to one carbonyl group (*δ*_C_ 172.8), six aromatic carbons (*δ*_C_ 132.7, 127.8, 124.5, 124.2, 124.0, 121.2), one quaternary carbon (*δ*_C_ 93.6), one oxygen-bearing methine (*δ*_C_ 65.4), three methylenes (*δ*_C_ 63.0, 30.6, 25.6), and one methyl (*δ*_C_ 16.3).

The planar structure was assigned by 2D NMR spectra (^1^H-^1^H COSY, HSQC, HMBC) (Figure 2). Analysis of the COSY spectrum suggested the presence of three isolated proton spin systems, including a 1,2-disubstituted aromatic system (–CH–CH–CH–CH–) and two ethyl fragments (–CH_2_–CH– and –CH_2_–CH_2_–). The key HMBC correlations from H-7 to C-8, C-9, C-13, from H-12 to C-13, and the chemical shift (*δ*_C_ 93.6, C-5; 63.2 C-7; 124.0 C-13) indicated the presence of dihydro-2*H*-benzo[d][1,3]oxazine structure. At the same time, HMBC correlations from H-3 to C-2 and C-5, from H-4 to C-2 and C-5 constructed a pyrrolidin-2-one moiety. The remaining ethan-1-hydroxytehyl group was connected to C-5 based on the HMBC correlation of H-2′ with C-1′ and C-5. The absolute configuration of **1** was assigned as (5*R*,1′*R*) by a single-crystal X-ray diffraction experiment using Cu-Kα radiation [16] (Figure 3) (Flack parameter = 0.01(4)).

Nigrosporine B (**2**) was obtained as a colorless oil, and its molecular formula was identified as the same as **1** with C_13_H_15_NO_3_ on the basis of the positive HR-ESIMS ions at *m*/*z* 256.0947 [M + Na]^+^ (calculated for 256.0950, C_13_H_15_NO_3_Na). Compound **2** shared the same planar structure as **1** and was further identified by 2D NMR spectra, including ^1^H-^1^H COSY, HSQC, and HMBC (Figure 2). The minor chemical shift variation in C-4 (*δ*_C_ 25.6, *δ*_H_ 2.62, 1.98 for **1**; *δ*_C_ 25.5, *δ*_H_ 2.68, 1.93 for **2**), C-5 (*δ*_C_ 93.6 for **1**; *δ*_C_ 94.0 for **2**), C-1′ (*δ*_C_ 65.4, *δ*_H_ 4.22 for **1**; *δ*_C_ 65.8, *δ*_H_ 4.19 for **2**), and C-2′ (*δ*_C_ 16.3, *δ*_H_ 1.13 for **1**; *δ*_C_ 16.3, *δ*_H_ 1.31 for **2**) were observed in the same CDCl_3_ solvent, which suggested that **2** could be a 5-epimer of **1** (Table 1). The absolute configuration of the secondary hydroxylated carbon (C-1′) was determined using a modified Mosher’s method [17,18,19,20]. The chemical shifts for H-2′, H-4, and H-7 of **1a** and **1b** were measured as *δ*_H_ 1.49, 4.94, 2.38 for **1a**, and *δ*_H_ 1.44, 4.96, 2.49 for **1b**, respectively. The observed differences in chemical shifts (Δ*δ* = *δ*_S_ − *δ*_R_) (Figure 4) suggested that the C-1′ absolute configuration is *R* the same as that of **1**. The ECD spectra of each conformer were calculated by a quantum chemical method at the [PBE1PBE/TZVP] level, and the Boltzmann-weighted ECD spectrum of (5*S*,1′*R*)-**2** was in good agreement with the experimental one (Figure 5). Thus, the absolute configuration of **2** was assigned as 5*S*,1′*R*.

Nigrosporine C (**3**) was obtained as a brown crystal. The molecular formula was determined as C_13_H_13_NO_3_ on the basis of the positive HR-ESIMS ions at *m*/*z* 254.0791 [M + Na]^+^ (calculated for 254.0730, C_13_H_13_NO_3_Na), implying eight degrees of unsaturation. Detailed analysis of its NMR spectroscopic data (Table 1) suggested that **3** was similar to the chemical structure of **1**, except for the presence of an additional signal attributed to a ketone group (*δ*_C_ 204.3) and the absence of oxygen-bearing methine (*δ*_C_ 65.4) in **3**. The key HMBC correlations from H-2′ and H-4 to C-1′ were allowed to assign the location of the ketone group in **3**. Compound **3** was crystallized from methanol by a slow evaporation method, yielding crystals in the monoclinic *P*2_1_/c space group. This structure was definitively confirmed through X-ray crystallography, as shown in Figure 6. The *P*2_1_/c space group of the X-ray structure indicated that compound **3** likely existed as enantiomers within the solid-state structure, which was consistent with the result of no signal in the CD spectrum and no optical activity sign in methanol. Subsequently, compound (±)-**3** was purified using chiral HPLC, leading to the isolation of the two enantiomers, (+)-**3** (*t*_R_ = 15.98 min) and (−)-**3** (*t*_R_ = 17.46 min), respectively. These enantiomers exhibited opposite Cotton effects in their CD spectra and opposite optical rotations (Figure 7). The calculated ECD data of (*R*)-**3** were in agreement with the experimental one of (+)-**3**, suggesting that the absolute configuration of (+)-**3** was *R,* and (−)-**3** was *S*. Therefore, (+)-**3** and (–)-**3** were named (+)-nigrosporine C and (−)-nigrosporine C, respectively.

The known isolates were identified as 2-(4-hydroxyphenyl) -2-aminoethanol (**4**) [21], *N*-(2-hydroxyphenyl)-acetamide (**5**) [22], *N*-phenethylacetamide (**6**) [23], 3-phenylpropane-2,3-diol (**7**) [24], 3-phenylpropane-1,2-diol (**8**) [25], ethyl 4-(2,3-dihydroxy -3- methylbutoxy) benzoate (**9**) [26], (1*S*,3*R*,4*S*,7*S*)-3,4-dihydroxy-α-bisabolol (**10**) [27], and schizostatin(**11**) [28] by comparison of their spectroscopic data with those reported in the literature.

Compounds **1**–**11** were evaluated for their anti-neuroinflammatory and AChE inhibitory activities. Compound **2** exhibited moderate inhibition on NO accumulation induced by LPS on BV-2 cells in a dose-dependent manner at 20 (47.2 ± 0.2 ng/mL), 50 (41.7 ± 0.4 ng/mL), and 100 (21.8 ± 0.1 ng/mL) μmol/L and without cytotoxicity in a concentration of 100.0 μmol/L (Figure 8). Simultaneously, compound **2** also showed moderate AChE inhibitory activities with IC_50_ values of 103.7 μmol/L(positive control, Donepezil, IC_50_ = 0.48 μmol/L). Additionally, all compounds were also evaluated for their antioxidant activities, and compound **5** displayed moderate antioxidant activity with EC_50_ values of 167.0 μmol/L(positive control, Vitamin C, EC_50_ = 146.0 μmol/L). Other compounds showed weak antioxidant (EC_50_ > 200.0 μmol/L) and AChE inhibitory activities (IC_50_ > 200.0 μmol/L).

## 3. Materials and Methods

### 3.1. General Experimental Procedures

Optical rotations were carried out on an MCP-200 polarimeter (Anton Paar, Graz, Austria) with MeOH as solvent at 25 °C. UV spectra were acquired on a Blue Star A spectrophotometer. IR data were performed on a Fourier transformation infrared spectrometer coupled with an EQUINOX 55 infrared microscope (Bruker, Fällanden, Switzerland). Here, 1D and 2D NMR spectra were tested on a Bruker Avance 400 MHz spectrometer (Bruker, Fällanden, Switzerland) using TMS as an internal standard. HR-ESIMS and ESIMS data were recorded on an LTQ-Orbitrap LC-MS spectrometer (Thermo Corporation, Waltham, MA, USA) and an ACQUITY QDA (Waters Corporation, Milford, MA, USA), respectively. HPLC preparative separations were performed on a Shimadzu Essentia LC-16. The Welch-Ultimate XB-C18 column (250 × 21.2 mm, 5 μm, 12 nm, Welch Materials, Inc., Shanghai, China) was used for preparative HPLC. Semi-preparative HPLC separations were performed on ACE-5-C18-AR and ACE-5-CN-ES columns (250 × 10 mm, 5 μm, 12 nm, Advanced Chromatography Technologies Ltd., Guangzhou, China). The silica gel (200–300 mesh, Qingdao Marine Chemical Inc., Qingdao, China) and Sephadex LH-20 (Amersham Biosciences, Uppsala, Sweden) were subjected to column chromatography (CC).

### 3.2. Fungal Material

The fungal strain *Nigrospora oryzae* SYSU-MS0024 was isolated from sediment, which was collected from the Zhuhai Mangrove Special Protected Area, Guangdong Province, China, in December 2021. This strain was identified by the sequence analysis of the rDNA ITS (internal transcribed spacer) region [29]. The sequence data of the fungal strain have been deposited at GenBank with accession no. PP422138. A BLAST search result suggested that the sequence was most similar (99.81%) to the sequence of *Nigrospora oryzae* SYSU-MS0024 (compared to MZ151376.1). The strain was preserved at the School of Marine Sciences, Sun Yat-Sen University.

### 3.3. Extraction and Isolation

The strain *Nigrospora oryzae* SYSU-MS0024 was grown on a solid-rice medium in a 1000 mL culture flask containing 50 g of rice and 50 mL of 3% artificial seawater after sterilization. A total of 120 flasks of fungal incubation were cultivated at room temperature for 30 days. The solid fermented substrate was extracted with MeOH four times to obtain a crude extract, then dissolved in H_2_O, and continuously extracted four times with EtOAc solvent. The EtOAc extract (30 g) was subjected to a silica gel column eluting with linear gradient petroleum ether/EtOAc (from 8:2 to 0:1) to obtain six fractions (A–F). Fr.B was chromatographed on a Sephadex LH-20 column with MeOH/CH_2_Cl_2_ (50:50) to afford four fractions (Fr.B.1 to Fr.B.4) and **11**. Fr.B.2 was separated by a silica gel column eluting with CH_2_Cl_2_/MeOH (97:3) to afford five fractions (Fr.B.2.1 to Fr.B.2.5). Fr.B.2.3 was further subjected to RP-HPLC (MeCN/H_2_O, 32:68 flow rate 2 mL/min, Aglient-XB-C18 column 10 × 250 mm, 5 μm) to afford **2** (3.7 mg, *t*_R_ = 13.4 min) and **9** (2.8 mg, t*_R_* = 17.6 min). Fr.C was chromatographed on a Sephadex LH-20 column with MeOH/CH_2_Cl_2_ (50:50) to afford three fractions (Fr.C.1 to Fr.C.3). Fr.C.2 was separated by a silica gel column eluting with CH_2_Cl_2_/MeOH (88:12) to afford four fractions (Fr.C.2.1 to Fr.C.2.4). Fr.C.2.2 was further subjected to RP-HPLC (MeCN/H_2_O, 35:65 flow rate 2 mL/min, Aglient-XB-C18 column 10 × 250 mm, 5 μm) to afford **6** (3.6 mg, *t*_R_ = 18.4 min) and **7** (3.4 mg, *t*_R_ = 21.5 min). Fr.C.2.3 was further subjected to RP-HPLC (MeCN/H_2_O, 33:67 flow rate 2 mL/min, Aglient-XB-C18 column 10 × 250 mm, 5 μm) to afford **1** (4.2 mg, *t*_R_ = 15.4 min) and **3** (2.8 mg, *t*_R_ = 15.9 min) and **8** (3.6 mg, *t*_R_ = 13.4). Fr.D was further purified by silica gel column to three fractions (Fr.D.1 to Fr.D.3), and Fr.D.2 was chromatographed on a Sephadex LH-20 column with MeOH/CH_2_Cl_2_ (50:50) to afford four fractions (Fr.D.2.1 to Fr.D.2.4). Fr.D.2.3 was further subjected to RP-HPLC (MeCN/H_2_O, 30:70 flow rate 2 mL/min, XB-C18 column 10 × 250 mm, 5 μm) to afford 4 (3.8 mg, *t*_R_ = 17.2 min) and **5** (2.8 mg, *t*_R_ = 21.4 min). Compound **10** (6.5 mg, *t*_R_ = 12.7 min) was obtained from Fr.D.2.3 by silica gel CC (CH_2_Cl_2_/MeOH, 100:1, 80:1, 60:1, 40:1, 20:1, 10:1 v/v) with a final purification on an ACE-C18-PFP column (Welch column, MeCN/H_2_O, 24:76 flow rate 2 mL/min). The enantiomeric mixture of **3** (2.8 mg) was applied on a chiral HPLC (70% n-hexane/isopropyl alcohol, flow rate 3 mL/min, NanoChrom UniChiral^®^ OD (5 μm, 10 × 250 mm) to yield (+)-**3** (*t*_R_ = 15.98 min, 1.0 mg) and (−)-**3** (*t*_R_ = 17.46 min, 1.1 mg) using the method described previously [30].

Nigrosporine A (**1**): colorless crystal; mp 136–137 °C; ${\left[\alpha \right]}_{D}^{20}$ +13.0 (*c* 1.00, MeOH); UV (MeOH) *λ*_max_ (log *ε*) 208 (3.81), 242 (3.52) nm; CD Δε (MeOH) *λ*_max_ (Δ*ε*) 216 (−13), 246 (13) nm; IR (neat) *ν*_max_ 3365, 2937, 2370, 1673, 1494, 1457, 1394, 1285, 1200 cm^−1^; ^1^H and ^13^C NMR data, see Table 1; HR-ESIMS *m*/*z* 256.0945 [M + Na]^+^ (calculated for C_13_H_15_NO_3_Na, 256.0950).

Nigrosporine B (**2**): brown oil;${\left[\alpha \right]}_{D}^{20}$ −13.7 (*c* 1.00, MeOH); UV (MeOH) *λ*_max_ (log *ε*) 208 (3.82), 243 (3.52)nm; CD Δε (MeOH) *λ*_max_ (Δ*ε*) 216 (13), 246 (−13) nm; IR (neat) *ν*_max_ 3377, 2926, 2855, 1602, 1491, 1390, 1196, 1073 cm^−1^; ^1^H and ^13^C NMR data, see Table 1; HR-ESIMS *m*/*z* 256.0947 [M + Na]^+^ (calculated for C_13_H_15_NO_3_Na, 256.0950).

Nigrosporine C (**3**): colorless crystal; mp 122–123 °C; ${\left[\alpha \right]}_{D}^{20}$ −2.1 (*c* 1.00, MeOH); UV (MeOH) *λ*_max_ (log *ε*) 206 (3.85), 241 (3.53)nm; IR (neat) *ν*_max_ 3369, 2922, 1703, 1606, 1491, 1454, 1368, 1193 cm^−1^; ^1^H and ^13^C NMR data, see Table 1; HR-ESIMS *m*/*z* 254.0791 [M + Na]^+^ (calculated for C_13_H_13_NO_3_Na, 254.0730).

(+)-(**3**): ${\left[\alpha \right]}_{D}^{20}$ +25.2 (c 0.12, MeOH); ECD (MeOH) *λ*_max_ (∆ε): 223 (-2.1), 248 (+8.5) nm.

(−)-(**3**): ${\left[\alpha \right]}_{D}^{20}$ −21.6 (c 0.12, MeOH); ECD (MeOH) *λ*_max_ (∆ε): 223 (+1.9), 249 (−8.0) nm.

### 3.4. Preparation of (S)-MTPA Ester and (R)-MTPA Ester

Compound **2** (1.0 mg) was dissolved in pyridine-*d*_5_ (0.5 mL) in an NMR tube, and then (*R*)-MPTACl (5.0 μL) was added to react at room temperature for 24 h. Then, the ^1^H NMR spectrum of the (*S*)-MTPA ester derivative (**2a**) was measured directly on the reaction mixture [17] Similarly, another reaction of **2** (1.0 mg), (S)-MPTACl(5.0 μL), and pyridine-*d*_5_ (0.5 mL) was performed, as described above for **1a** to afford **2b**.

(*S*)-MTPA Ester of (**2a**): ^1^H NMR (Appendix A) (selected signals, pyridine-*d*_5_, 400 MHz) *δ*_H_: 4.94 (2H, m, H-7), 2.38 (1H, m, H-4a), 1.955 (1H, m, H-4b), 1.49 (3H, d, H-2′).

(*R*)-MTPA Ester of (**2b**): ^1^H NMR (Appendix A) (selected signals, pyridine-*d*_5_, 400 MHz) *δ*_H_: 4.96 (2H, m, H-7), 2.49 (1H, m, H-4a), 1.959 (1H, m, H-4b), 1.44 (3H, d, H-2′).

### 3.5. X-ray Crystallographic Analysis

Compounds **1** and **3** were obtained as colorless crystals using the vapor diffusion method. The single crystal X-ray diffraction data were recorded on a Rigaku Oxford Diffraction with Cu-Kα radiation (λ = 1.54178A). The structures were solved by direct methods (SHELXS-97 and Olex2-1.2) and refined using full-matrix least-square difference Fourier techniques [31]. Crystallographic data for **1** and **3** have been deposited with the Cambridge Crystallographic Data Centre. Copies of the data can be obtained, free of charge, on application to the Director, CCDC, 12 Union Road, Cambridge CB2 1EZ, UK (fax: 44-(0)1223-336033, or e-mail: deposit@ccdc.cam.ac.uk).

Compound **1**: C_13_H_15_NO_3_ (*Mr* = 233.26 g/mol), monoclinic, space group *P*2_1_, *a* = 8.75680(10) Å, *b* = 6.60250(10) Å, *c* = 10.50300(10) Å, *α* = 90°, *β* = 103.9630(10)°, *γ* = 90°, *V* = 589.306(13) Å^3^, *Z* = 2, *T* = 306.14(10) K, *µ*(Cu Kα) = 1.168 mm^−1^, *D*_calc_ = 1.430 g/cm^3^; 2289 reflections measured (*R*_int_ = 0.0183, *R*_sigma_ = 0.0124), which were used in all calculations. The final *R*_1_ was 0.0440 (*I* ≥ 2*u*(*I*)), and *wR*_2_ was 0.1018. The Flack parameter was 0.01(4). The goodness of fit on *F*^2^ was 1.198. CCDC 2324841.

Compound **3**: C_13_H_13_NO_3_ (*Mr* = 231.24 g/mol), monoclinic, space group *P*2_1_/c, *a* = 8.9565(4) Å, *b* = 8.7773(3) Å, *c* = 14.4823(5) Å, *α* = 90°, *β* = 100.771(4)°, *γ* = 90°, *V* = 1118.45(7) Å^3^, *Z* = 4, *T* = 306.14(10) K, *µ*(Cu Kα) = 0.809 mm^−1^, *D*_calc_ = 1.373 g/cm^3^; 2297 reflections were measured (*R*_int_ = 0.0427, *R*_sigma_ = 0.0271), which were used in all calculations. The final *R*_1_ was 0.0674 (*I* ≥ 2*u*(*I*)), and *wR*_2_ was 0.1860. The Flack parameter was 0.01(4). The goodness of fit on *F*^2^ was 1.136. CCDC 2336893.

### 3.6. Calculation of the ECD Spectra

The Merck molecular force field (MMFF) and density functional theory (DFT) calculations were performed using Spartan’14 software from Wavefunction Inc. (Irvine, CA, USA) and the Gaussian 09 program, respectively [32]. Conformers were subjected to DFT geometry optimizations at the B3LYP/def2-SVP level with the solvation model PCM for methanol. Frequency calculations were carried out at the same level. The optimized low-energy conformers were subjected to single-point energy calculations using the DFT method at the M06-2X/def2-TZVP/SDM(MeOH) level of theory. Fifty excited-state rotatory strengths were computed. The population of each conformer was determined using a Boltzmann distribution with Gibbs free energy calculated by Shermo [33]. To simulate ECD spectra, the optimized low-energy conformers were subjected to the TDDFT calculations at the PBE1PBE/CAM-B3YLP/TZVP level in methanol. The ECD spectrum was generated by the program SpecDis [34]. All calculations were performed by Tianhe-2 in the National Super Computer Center in Guangzhou.

### 3.7. Anti-Inflammatory Activity

NO determination: all compounds were examined for the effect on the release of NO by BV-2 cell lines using the method described previously [35]. All experiments were carried out in five replicates. Microglia Viability Assay: The effects of compounds **1**–**11** on the viability of BV2 microglia were carried out by MTT assay. BV2 microglia were first seeded at a density of 4000 cells per well for 12 or 24 h using a 96-well plate and then treated with compounds at the concentrations of 1– 100 μmol/L for another 36 h. After that, cells were incubated with MTT (0.5 mg/mL) in a 5% CO_2_ incubator at 37 °C. Three hours later, the medium was removed, and DMSO (150 μL) was added into each well to dissolve formazan crystals. The fluorescent absorbances were measured at 570 nm using an ELISA microplate reader (Biotek, Winooski, VT, USA).

### 3.8. AChE Inhibitory Activity Assay

All compounds were examined for inhibitory activity against AChE using an optimized colorimetric method described previously [36]. All tests were carried out in three replicates. Donepezil hydrochloride was taken as the positive control.

### 3.9. DPPH Free Radical Scavenging Assay

An optimized colorimetric method described previously was used to evaluate the DPPH free radical scavenging activities of compounds **1**–**11** [37]. All tests were measured in three replicates. Vitamin C was taken as a positive control.

## 4. Conclusions

In summary, three new alkaloids (**1**–**3**), together with eight known compounds (**4**–**11**), were separated from the mangrove fungus *Nigrospora oryzae* SYSU-MS0024. The stereo structures of compounds **1**–**3** were deduced by a combination of electronic circular dichroism (ECD) calculations, Mosher’s method, and X-ray single crystal diffraction technique using Cu Kα radiation. In bioassays, compound **2** showed moderate inhibition of LPS-induced NO production on BV-2 cells in a dose-dependent manner at 20, 50, and 100 μmol/L and without cytotoxicity in a concentration of 100 μmol/L. Simultaneously, compound **2** also showed moderate AChE inhibitory activities with IC_50_ values of 103.7 μmol/L. Additionally, compound **5** displayed moderate antioxidant activity with EC_50_ values of 167.0 μmol/L.

## Figures and Tables

**Figure 1 marinedrugs-22-00214-f001:**
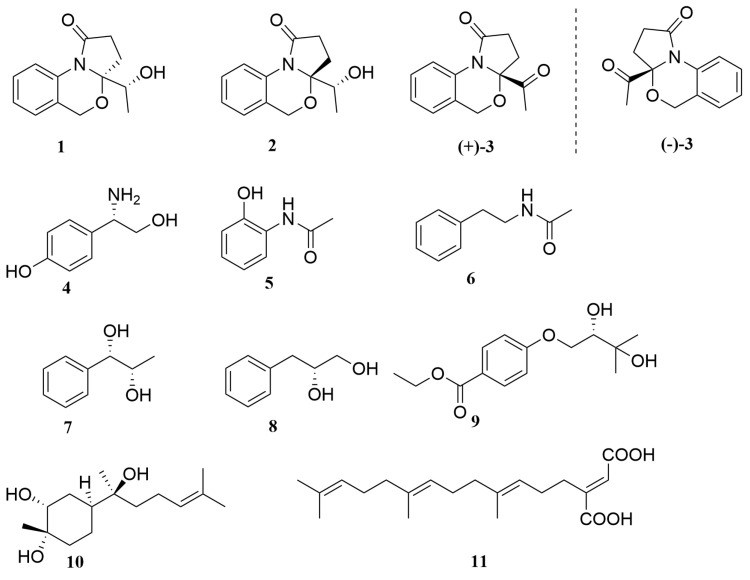
Chemical structures of compounds **1**–**11**.

**Figure 2 marinedrugs-22-00214-f002:**
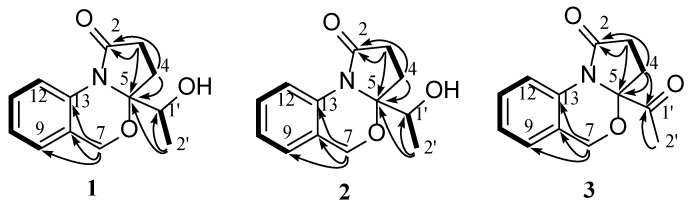
Key ^1^H-^1^H COSY (black line) and HMBC (black arrow) correlations of compounds **1**–**3**.

**Figure 3 marinedrugs-22-00214-f003:**
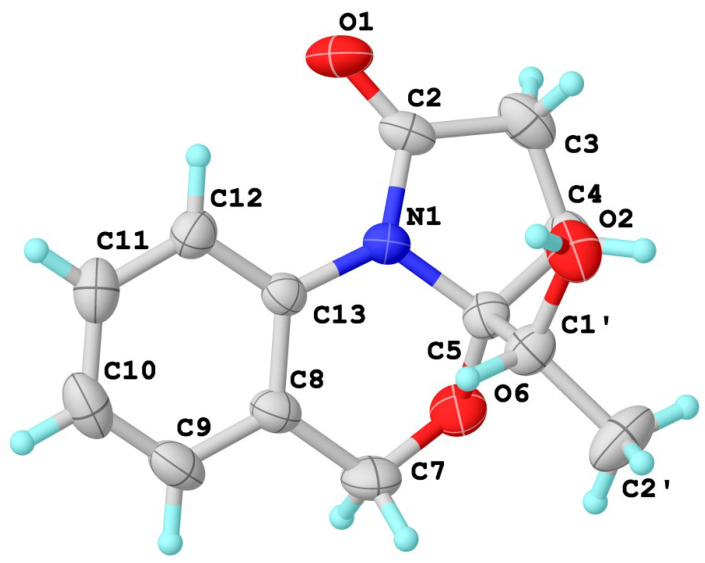
X-ray crystallographic analysis of **1**.

**Figure 4 marinedrugs-22-00214-f004:**
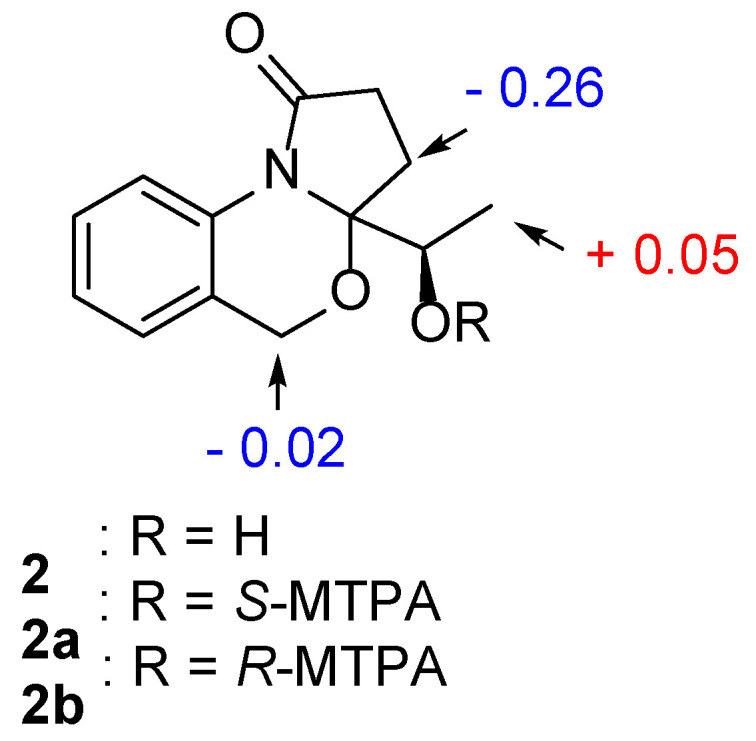
∆*δ* = *δ*_S_ − *δ*_R_ values in ppm obtained from the MTPA esters of **2**.

**Figure 5 marinedrugs-22-00214-f005:**
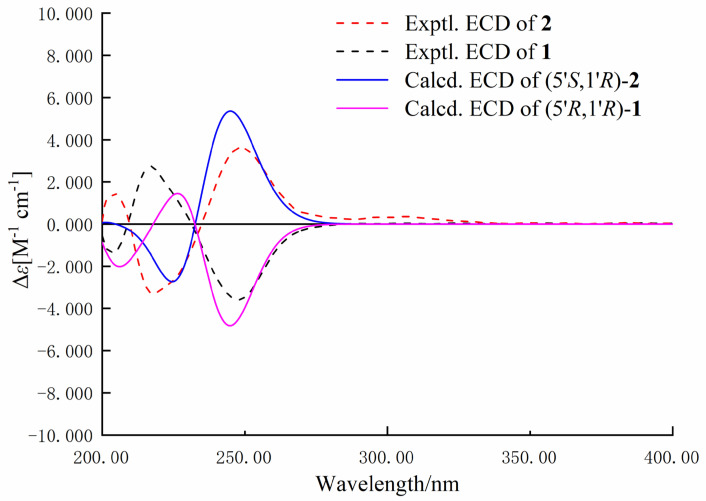
Experimental and calculated ECD spectra of compounds **1** and **2** (in MeOH).

**Figure 6 marinedrugs-22-00214-f006:**
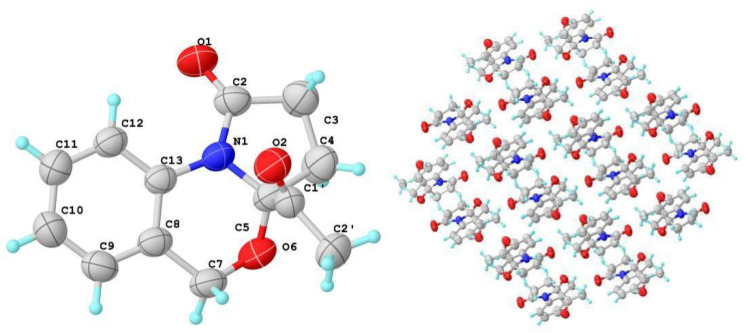
X-ray crystallographic analysis and molecular packing properties of **3**.

**Figure 7 marinedrugs-22-00214-f007:**
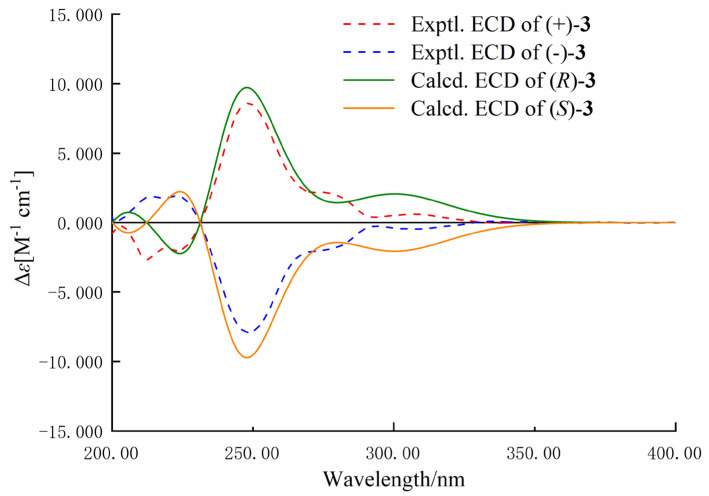
Experimental and calculated ECD spectra of compound **3** (in MeOH).

**Figure 8 marinedrugs-22-00214-f008:**
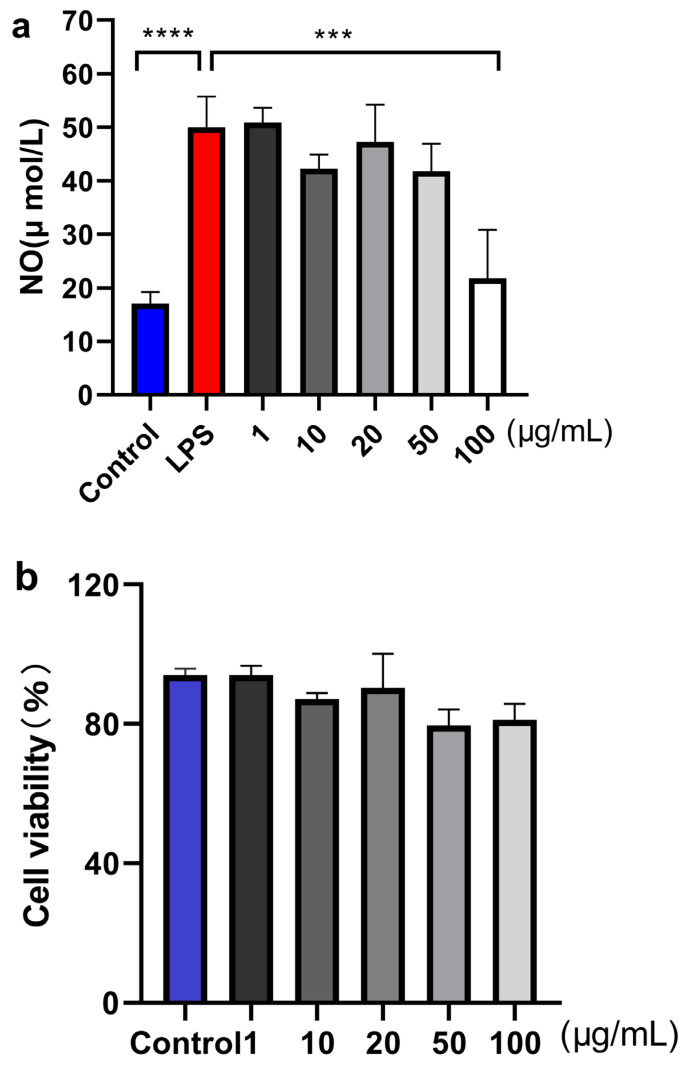
Effects of compound **2** on LPS-induced NO production in BV2 microglia (**a**), Effects of different concentrations of compound **2** on BV-2 cells viability (**b**). Mean ± SD of five replicates is shown. **** *p* < 0.0001, LPS group versus control group; *** *p* < 0.001, sample group versus LPS group.

**Table 1 marinedrugs-22-00214-t001:** ^1^H and ^13^C NMR spectroscopic data of **1**–**3** (CDCl_3_).

No.	1		2		3	
	*δ*_C_, Type	*δ*_H_, Mult (*J* in Hz)	*δ*_C_, Type	*δ*_H_, Mult (*J* in Hz)	*δ*_C_, Type	*δ*_H_, Mult (*J* in Hz)
2	172.8, C		174.0, C		172.2, C	
3	30.6, CH_2_	2.62, m	30.9, CH_2_	2.60, m	28.6, CH_2_	2.62, m
4	25.6, CH_2_	2.62, m; 1.98,m	25.5, CH_2_	2.68, m; 1.93,m	29.4, CH_2_	2.62, m; 2.30, m
5	93.6, C		94.0, C		94.8, C	
7	63.0, CH_2_	5.04, d, (15.6); 4.92, d, (15.6)	63.3, CH_2_	4.89, d, (15.6); 4.84, d, (15.6)	65.1, CH_2_	4.95, s
8	132.7, C		132.5, C			
9	127.8, CH	7.30, t, (6.6)	127.9, CH	7.30, t, (6.6)	128.0, CH	7.30, t, (6.6)
10	124.5, CH	7.14, td, (7.5, 1.2)	124.7, CH	7.1, td, (7.5, 1.2)	124.9, CH	7.11, td, (7.5, 1.2)
11	124.2, CH	7.05, d, (7.8)	124.3, CH	7.04, d, (7.8)	124.3, CH	6.98, d, (7.6)
12	121.2, CH	8.14, d, (8.4)	121.7, CH	8.18, d, (7.1)	120.5, CH	8.27, d, (8.2)
13	124.0, C		124.1, C		122.9, C	
1′	65.4, CH	4.22, qd, (6.4, 1.1)	65.8, CH	4.19, q, (6.4)	204.3, C	
2′	16.3, CH_3_	1.13, d, (6.4)	16.3, CH_3_	1.31, d, (6.4)	23.9, CH_3_	2.23, s

## Data Availability

All data are provided in full in the results section and Appendix A of this paper.

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
