# Peer review of "Bioactive Alkaloids from the Mangrove-Derived Fungus *Nigrospora oryzae* SYSU-MS0024"

_marinedrugs, 2024, doi:10.3390/md22050214_

Round 1
Reviewer 1 Report
Comments and Suggestions for Authors
This paper reports the isolation, structure elucidation, and the biassay results of threee new and eight known compounds, from the marine fungus Nigrospora oryzae SYSU-MS0024.
Overall, the structure elucidation is well addressed. COSY, HSQC and HMBC are well explained. X-Ray is key to solve compounds 1 and 3. However some minor issues should be corrected.
First of all compounds 1 and 2 are wrongly drawn. They should be pictured as it is shown in the attached file.
Compound 1: Discussion of the 1H NMR of the aromatic fragment. The coupling constants make no sense. Please average the J values for ortho-couplings and meta-couplings protons. Pay attention to the signal at 7.30 (t 6.6 Hz). It should be measured properly.
The calculated exact masas for the (M+Na)+ ion should be 256.0944 instead of 256.0950. The formula is positive-charged (C13H15NaO3)+, therefore the mass of one electron should be removed. Same for compound 2, and in the experimental part. This issue must be revised in all molecular formulae along the paper.
How many electronic states were used in the ECD calculations? Please add them in the experimental part. In the discussion add the level of theory used in ECD DFT calculation (end on page 4: the predicted ECD spectrum at PBE1PBE/CAM-B3LYP/TZP level....)
Please, fix table 1 for compound 2 (13C).
On page 3, line 8 after table 1... The remaining ethan-1-ol group.... the remaining 1-hydroxytehyl group...

Author Response
Q1: First of all compounds 1 and 2 are wrongly drawn. They should be pictured as it is shown in the attached file.
Response: Thank you for your suggestion. Compounds 1 and 2 have been redrawn as suggested in the manuscript.
Q2: Compound 1: Discussion of the 1H NMR of the aromatic fragment. The coupling constants make no sense. Please average the J values for ortho-couplings and meta-couplings protons. Pay attention to the signal at 7.30 (t 6.6 Hz). It should be measured properly.
Response: Thank you for your suggestion. In theory, the four protons on the 1,2-disubstituted aromatic ring should be split into dd, td, td, and dd, respectively. However, due to instruments resolution or other factors, in fact we most often can not obtain such an ideal cleavage for it will overlap. In our case, the splitting of protons at δH 7.30 overlap so that we can only see a t-peak.
Q3: The calculated exact mass for the (M+Na) ion should be 256.0944 instead of 256.0950. The formula is positive-charged (C+13H15NaO3)+, therefore the mass of one electron should be removed. Same for compound 2, and in the experimental part. This issue must be revised in all molecular formulae along the paper.
Response: Thank you for your suggestion. Indeed, we have a positive ion mode, but it's a sodium peak, not a hydrogenation peak, so the mass doesn’t need to removed one electron.
Q4: How many electronic states were used in the ECD calculations? Please add them in the experimental part. In the discussion add the level of theory used in ECD DFT calculation (end on page 4: the predicted ECD spectrum at PBE1PBE/CAM-B3LYP/TZP level....)
Response: Thank you for your suggestion. 50 electronic states were used in the ECD calculations, which was added in the experimental part. In the discussion, we add PBE1PBE/TZVP level of theory used in ECD DFT calculation for compound 2.
Q5: I Please, fix table 1 for compound 2 (13C).
Response: Thank you for your suggestion. We have fixed table 1 for compound 2 as suggested in the manuscript.
Q6: On page 3, line 8 after table 1... The remaining ethan-1-ol group.... the remaining 1-hydroxytehyl group...
Response: Thank you for your suggestion. We have revised “The remaining ethan-1-ol group.” to “the remaining 1-hydroxytehyl group” as suggested.
Reviewer 2 Report
Comments and Suggestions for Authors
The manuscript submitted by Chen and coworkers describes the isolation of several secondary metabolites from Mangrove-derived Fungus Nigrospora oryzae. The isolation procedure and structure elucidation allowed the identification of two novel compounds as pure enantiomer and a racemic mixture of a third new metabolite. Moreover, the authors have applied the Mosher method to assign the absolute configuration of compounds 1 and 2 by the double derivatization protocol. The isolated new and known metabolites were also evaluated for their anti-neuroinflammatory and AChE inhibitory activities with interesting results.
The major point of criticism regards the interpretation of the data obtained by Mosher’s method. In particular, the authors have described the implemented method of Mosher of carbinol absolute configuration assignment. It is correct but why did not they include into DdH calculation also the methylene at C3? The same concept regards the aromatic CH since the anisotropic effects of MTPA are favored on aromatic as well as rigid scaffold. Indeed, such rigid conformers are ideal structure for the application of this kind of method. For example, by a cursory search on PubMed it is easy to read recent articles on stereochemical assignment (published by the same group) of structures with few protons and many quaternary carbons (10.3390/molecules29071598; 10.3390/molecules28010057). The effects of chiral auxiliary agents (such as MTPA or MPA) are evident on the entire skeleton of the molecules.
These data should be added and modified by the authors to reinforce their approach considering that these are new metabolites in the literature, and they associated ECD calculation with chemical synthesis. The literature should be implemented, too.
Additionally, it is quite rare to find a DdH difference on the third decimal, are you sure?
This major points should be addressed and after that the paper could be accepted for publication.
Comments on the Quality of English LanguageMinor editing of English language
Author Response
Q1: The major point of criticism regards the interpretation of the data obtained by Mosher’s method. In particular, the authors have described the implemented method of Mosher of carbinol absolute configuration assignment. It is correct but why did not they include into DdH calculation also the methylene at C3? The same concept regards the aromatic CH since the anisotropic effects of MTPA are favored on aromatic as well as rigid scaffold. Indeed, such rigid conformers are ideal structure for the application of this kind of method. For example, by a cursory search on PubMed it is easy to read recent articles on stereochemical assignment (published by the same group) of structures with few protons and many quaternary carbons (10.3390/molecules29071598; 10.3390/molecules28010057). The effects of chiral auxiliary agents (such as MTPA or MPA) are evident on the entire skeleton of the molecules. These data should be added and modified by the authors to reinforce their approach considering that these are new metabolites in the literature, and they associated ECD calculation with chemical synthesis. The literature should be implemented, too.
Response: Thank you for your suggestion. The absolute configuration of 2 was established by a convenient Mosher ester procedure in which the sample was treated with MTPA chlorides in deuterated pyridine directly in NMR tubes[1]. Therefore, the signals of aromatic CH protons of compound 2 were overlapped with deuterated pyridine, and the chemical shift of Δδ was not assigned. At the same time, the methylene protons at C-3 was overlapped, differences of chemical shifts was not assigned. According to the protocol of the Mosher ester analysis for the determination of absolute configuration of stereogenic (chiral) carbinol carbons, our experimental result could determine the absolute configuration of C-1’.
On the basin of the reviewer’s suggestion, we also added the reference[1-3] of recent articles on stereo-chemical assignment (published by the same group) of structures with few protons and many quaternary carbons (10.3390/molecules29071598; 10.3390/molecules28010057) for discussion in manuscript.
- Su, B.N.; Park, E. J.; Mbwambo, Z. H.; Santarsiero, B. D.; Mesecar, A. D.; Fong, H. H. S.; Pezzuto, J. M.; Kinghorn, A. D., New Chemical Constituents of Euphorbia quinquecostata and Absolute Configuration Assignment by a Convenient Mosher Ester Procedure Carried Out in NMR Tubes. Journal of natural products 2002,65, (9), 1278-1282.
- Sparaco, R.; Cinque, P.; Scognamiglio, A.; Corvino, A.; Caliendo, G.; Fiorino, F.; Magli, E.; Perissutti, E.; Santagada, V.; Severino, B., 3-Nitroatenolol: First Synthesis, Chiral Resolution and Enantiomers’ Absolute Configuration. Molecules 2024,29, (7), 1598.
- Sparaco, R.; Scognamiglio, A.; Corvino, A.; Caliendo, G.; Fiorino, F.; Magli, E.; Perissutti, E.; Santagada, V.; Severino, B.; Luciano, P., Synthesis, Chiral Resolution and Enantiomers Absolute Configuration of 4-Nitropropranolol and 7-Nitropropranolol. Molecules 2023,28, (1), 57.
Q2: Additionally, it is quite rare to find a DdH difference on the third decimal, are you sure?
Response: Thank you for your suggestion. We have revised the third decimal of ΔδH difference as the second decimal of that.
Round 2
Reviewer 2 Report
Comments and Suggestions for Authors
The authors clarified the major critism of this paper
Comments on the Quality of English LanguageMinor typological mistakes